# A Chiron approach towards the stereoselective synthesis of polyfluorinated carbohydrates

Vincent Denavit[1], Danny Lainé[1], Jacob St-Gelais[1], Paul A. Johnson[1] & Denis Giguère [1]

The replacement of hydroxyl groups by fluorine atoms on hexopyranose scaffolds may allow access to the discovery of new chemical entities possessing unique physical, chemical and ultimately even biological properties. The prospect of significant effects generated by such multiple and controlled substitutions encouraged us to develop diverse synthetic routes towards the stereoselective synthesis of polyfluorinated hexopyranoses, six of which are unprecedented. Hence, we report the synthesis of heavily fluorinated galactose, glucose, mannose, talose, allose, fucose, and galacturonic acid methyl ester using a Chiron approach from inexpensive levoglucosan. Structural analysis of single-crystal X-ray diffractions and NMR studies confirm the conservation of favored $^4C_1$ conformation for fluorinated carbohydrate analogs, while a slightly distorted conformation due to repulsive 1,3-diaxial F···F interaction is observed for the trifluorinated talose derivative. Finally, the relative stereochemistry of multi-vicinal fluorine atoms has a strong effect on the lipophilicities (log$P$).

[1] Département de chimie, 1045 av. De la Médecine, Université Laval PROTEO, RQRM, Québec City, QC G1V 0A6, Canada. Correspondence and requests for materials should be addressed to D.Gèr. (email: denis.giguere@chm.ulaval.ca)

The synthesis, physical characterizations, and biological investigations of fluorinated compounds have attracted tremendous research interest in the past decades.[1,2] In this context, the efficient and controlled incorporation of fluorine atoms into organic derivatives has quickly become a powerful tool to discover original chemical entities with unique physical, chemical and even biological properties.[3–5]

The replacement of hydroxyl groups with fluorine atoms to generate fluorine-substituted analogs of naturally occurring or biologically active organic compounds is extensively studied.[6] The rationale to prepare such compounds arises from similarities between OH group and F atom in regard to polarity and isosteric relationship. Another important feature is the loss of hydrogen donating capacity for the F atom, but the high C−F bond energy renders them resistant to metabolic transformation. Finally, the addition of a fluorine group can lead to greater lipophilicity, which in turn can increase bioavailability, tissue distribution and cell permeability.[7–9]

Since only a limited number of fluorine-containing natural products have been isolated so far,[10] chemical syntheses[11] or enzymatic transformations[12] represent the best routes to study and access complex organofluorines. The synthesis of fluorine-containing contiguous stereogenic center is challenging.[13] Despite significant progress in asymmetric fluorination methodology,[14–18] the development of innovative synthetic methods should attract more attention. Tremendous synthetic efforts greatly contributed to the presence of a large number of fluorinated agrochemicals and pharmaceuticals on the market and also useful biochemical probes for the in vivo magnetic resonance imaging.[19–21] Representative recent examples of polyfluorinated organic molecules are presented in Fig. 1a. The intriguing all-cis-1,2,3,4,5,6-hexafluorocyclohexane 1 represents one of the most striking candidate and the synthetic tour de force was achieved by the group of O'Hagan in 2015,[22] and more recently by the group of Glorius with a Rh-catalyzed single step from hexafluorobenzene.[23] All of the fluorine atoms are cis in this molecule and the high facial polarization generates an unusually large dipole moment. Among promising applications, the use of this resulting Janus-faced structure may open up new avenues in material science. In 2016, the group of Carreira presented the synthesis of Fluorodanicalipin A 2, a fluorinated analog of chlorosulfolipid danicalipin A.[24] This pioneering work shows that both bromo-danicalipin and fluoro-danicalipin A disclosed similar solution conformations, but adverse effects may be due to the lipophilicity of the halogens. This example clearly demonstrates that preparation of fluorinated analogs of natural products may provide compounds with novel biological activities.

Interestingly, numerous fluorinated carbohydrates were also widely investigated as interesting and suitable imaging agents for [18F]-positron-emitting tomography for cancer diagnosis,[25,26] in the footsteps of the early development of radiopharmaceutical [18F]FDG (2-deoxy-2-(18F)fluoro-D-glucose).[27] Fluorinated carbohydrates also play interesting roles in biological systems as mechanistic probes or to modulate lectin−carbohydrate interactions.[28–30] Almost two decades ago, the group of DiMagno synthesized the hexafluorinated analog 3[31] (Fig. 1a) and this compound crosses red blood cell membrane at a tenfold higher rate than glucose. This example indicates that increasing the polar hydrophobicity may be a useful strategy for improving biological molecular recognition.[32] This outcome has recently been proved accurate by the development of tetrafluoroethylene-containing monosaccharide 4 by the group of Linclau.[33,34] This compound gained affinity to UDP-galactopyranose mutase from *Mycobacterium tuberculosis* showing that tetrafluorination can have beneficial effect on binding compared to unmodified analogs.

This example strongly suggests that more research should be directed towards polyfluorination in the design of carbohydrate mimetics.

In this context, the stereoselective synthesis[35,36] and conformational analysis[37] of polyfluorinated derivatives, and *a fortiori* multi-vicinal organofluorine isomers, remain a tedious challenge. Consequently, systematic biological investigations of heavily fluorinated carbohydrates have often been impeded by the weak efficiency of the multistep sequences used in de novo approaches.[28–30,33,34,38,39] In addition, these long synthetic sequences generally give rise to enantiomeric or diastereomeric mixtures of products, leading to difficult silica gel chromatographies for isolation of pure products. Herein, we proposed a Chiron approach for the preparation of heavily fluorinated sugars shown in Fig. 1b. The accessible variety of synthesized derivatives spans over seven distinct members of functionalized carbohydrates containing distinctive cis and trans chemical relationships between fluorine atoms. More specifically, original polyfluorinated D-galactopyranoside 5, D-galacturonic acid methyl ester 6, and D-fucoside 7 were generated from this approach, and contained a 2,3-trans, 3,4-cis pattern for the integrated fluorine atoms. The versatility of the methodology was further demonstrated with the access to trifluorinated D-glucose derivative 8 (2,3-trans, 3,4-trans), along with D-mannose 9 (2,3-cis, 3,4-trans), D-talose 10 (2,3-cis, 3,4-cis), and D-allose 11 (2,3-cis, 3,4-cis). Beyond its versatility, this convenient methodology included practical features that allowed operations on a large scale starting from inexpensive starting material. The proposed sequences enable a minimal usage of protection/deprotection cycles, allow an excellent regio- and stereocontrols; and avoid tedious purifications. Our synthetic endeavors will start from commercially available 1,6-anhydro-β-D-glucopyranose (levoglucosan). This choice has been motivated since the 1,6-anhydro core prevents protection of both O-6 and anomeric positions and allows navigation on the pyran ring to install fluorine atoms using simple experimental protocols. Finally, the discovery of rather unique approach to construct multiple C−F bonds in one single transformation is described herein.

## Results

**Synthesis of 2,3,4,6-tetrafluorinated galactoside**. The synthesis of 2,3,4,6-tetradeoxy-2,3,4,6-tetrafluorogalactopyranoside 27 is described in Fig. 2 and was initiated with easily accessible Cerny's epoxide 13 preliminarily generated from levoglucosan 12 in an efficient 4-step sequence.[40] Nucleophilic fluorination of the 2,3-anhydro derivative 13 was subsequently achieved upon exposure to $KHF_2$ in 73% yield. Then treatment of resulting compound 14 with Deoxo-Fluor™ furnished 2,3-dideoxy-difluoroglucose 15 with complete retention of configuration (2,3-trans relationship).[41] A $TiCl_4$-mediated benzyl deprotection generated 16 containing the desired free hydroxyl group, which was further activated as triflate (17) and subjected to a nucleophilic fluorination using TBAF. Despite several attempts, 1,6-anhydro-2,3,4-trideoxy-2,3,4-trifluoro-β-D-galactopyranose 18 proved to be difficult to isolate due to its high volatility. Consequently, acetolysis of the crude mixture under acidic conditions ($H_2SO_4$, $Ac_2O$) furnished the desired di-acetylated derivative 19 in a satisfactory 63% yield over 3 steps (α/β = 5:1), with the anticipated 2,3-trans, 3,4-cis relationship ($^{19}F$ NMR (470 MHz, Chloroform-d): $^3J_{F2-H3}$ = 12.8 Hz, $^3J_{F4-H3}$ = $^3J_{F4-H5}$ = 27.0 Hz, for details, see Supplementary Fig. 14).[42] The last hurdle involved the C-6 fluorination and to that end, an O-aryl group was first installed to block the anomeric position. It is well documented that an electron-withdrawing polyfluoroalkyl group destabilizes adjacent carbocation center.[43–45] In this context, glycosylation

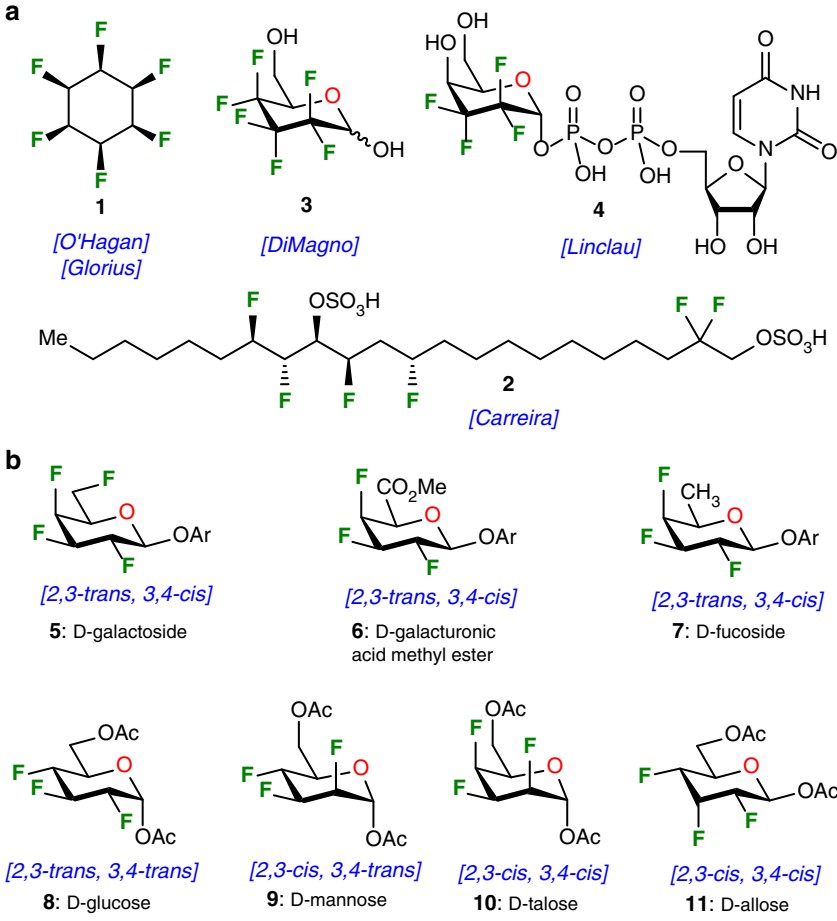

**Fig. 1** Heavily fluorinated organic molecules. **a** all-*cis*-1,2,3,4,5,6-hexafluorocyclohexane **1**, fluorodanicalipin A **2**; hexafluorinated carbohydrate analogs **3**, and tetrafluorinated UDP-galactopyranose **4**; **b** This work: fluorinated D-galactoside **5**, D-galacturonic acid methyl ester derivative **6**, D-fucoside **7**, D-glucose **8**, D-mannose **9**, D-talose **10**, and D-allose **11**

involving an oxocarbenium species was avoided and as a result, a phase-transfer-catalyzed nucleophilic displacement was established. The α-galactosyl bromide **20** was slowly generated (2 days) using an excess of hydrogen bromide in acetic acid from intermediate **19**. Then, intermediate **20** was treated with methyl *p*-hydroxybenzoate and as expected, the desired β-galactoside **21** was isolated in a 60% yield, along with the adverse elimination product that led to the trifluoro glycal derivative in 20% yield. The first C-6 deoxofluorination attempts were directed to the activation of C-6 hydroxyl group **22** as triflate **23** and subsequent treatment with TBAF generated within minutes a 10:1 mixture of elimination products: *arabino*-hex-5-enopyranoside **24** and compound **25** in 73% yield. Only trace amounts of the targeted tetrafluorinated product were observable under these conditions. We suspected that side product **24** was prone to elimination reactions, and this tendency was confirmed when its treatment with TBAF over a prolonged period of time (3 days) allowed a clean conversion to **25** in 86% yield. The second option proved to be more direct and successful. A DAST-mediated deoxofluorination on 2,3,4-trifluorinated galactopyranoside **22** smoothly generated 2,3,4,6-tetradeoxy-2,3,4,6-tetrafluorohexopyranoside **26** in 96% yield [as a 1:1 mixture with the chromatographically separable *arabino*-hex-5-enopyranoside derivative **24** (for details, see Supplementary Methods)]. In order to ease the recrystallization procedure, the benzoate aglycone was ultimately transformed into the corresponding carboxylic acid **27** with the use of aqueous 1 M LiOH solution.

**Conformational analysis and theoretical calculations**. X-ray analyses proved to be very instructive. In the solid state, compound **27** is a dimer (Fig. 2) and the pyran ring adopts the $^4C_1$ conformation. Interestingly, the 1,3-C−F bond repulsion usually noticed when 2 fluorine atoms are placed 1,3 on a hydrocarbon chain[46,47] is not observed in this particular case, even though the fluoromethyl group (C5−C6 linkage) is free to rotate. In the solid state, compound **27** adopts the highest energetic conformation (GG), which is unusual for galactoside derivatives.[48] At first, we proposed that this 1,3-alignment increases the overall molecular dipole moment[49,50] allowing intermolecular C−F⋯H−C interactions responsible, in part, for the solid-state ordering as seen in the crystal structures packing (Fig. 3a).[32,51,52] Intermolecular interactions include π-stacking from the aromatic portion and also possibly hydrogen bonds involving the C-6 fluorine atoms: $d_{H6,F6} = 2.79$ Å ($C_6−H⋯F_6$: $\theta = 121.4°$), $d_{H5,F6} = 2.63$ Å ($C_5−H⋯F_6$: $\theta = 111.9°$), and $d_{H4,F6} = 2.51$ Å ($C_4−H⋯F_6$: $\theta = 123.2°$).[53] Also, the C-4 fluorine might be involved in this hydrogen bonding network: $d_{H4,F4} = 2.86$ Å ($C_4−H⋯F_4$: $\theta = 118.3°$), and $d_{H3,F4} = 2.57$ Å ($C_3−H⋯F_4$: $\theta = 113.7°$), for details see Supplementary Table 17. One more argument supporting possible intermolecular C−F⋯H−C interactions is the shielded chemical shift of fluorine atoms for compound 27: $^{19}F$ NMR (470 MHz, Acetone-$d_6$) δ −201.83 (F3), −207.14 (F2), −217.20 (F4), −230.43 (F6) (for details, see Supplementary Fig. 53).[54] Density functional theory (DFT) calculations were performed with Gaussian 09, revision E.01[55] to evaluate our hypothesis.

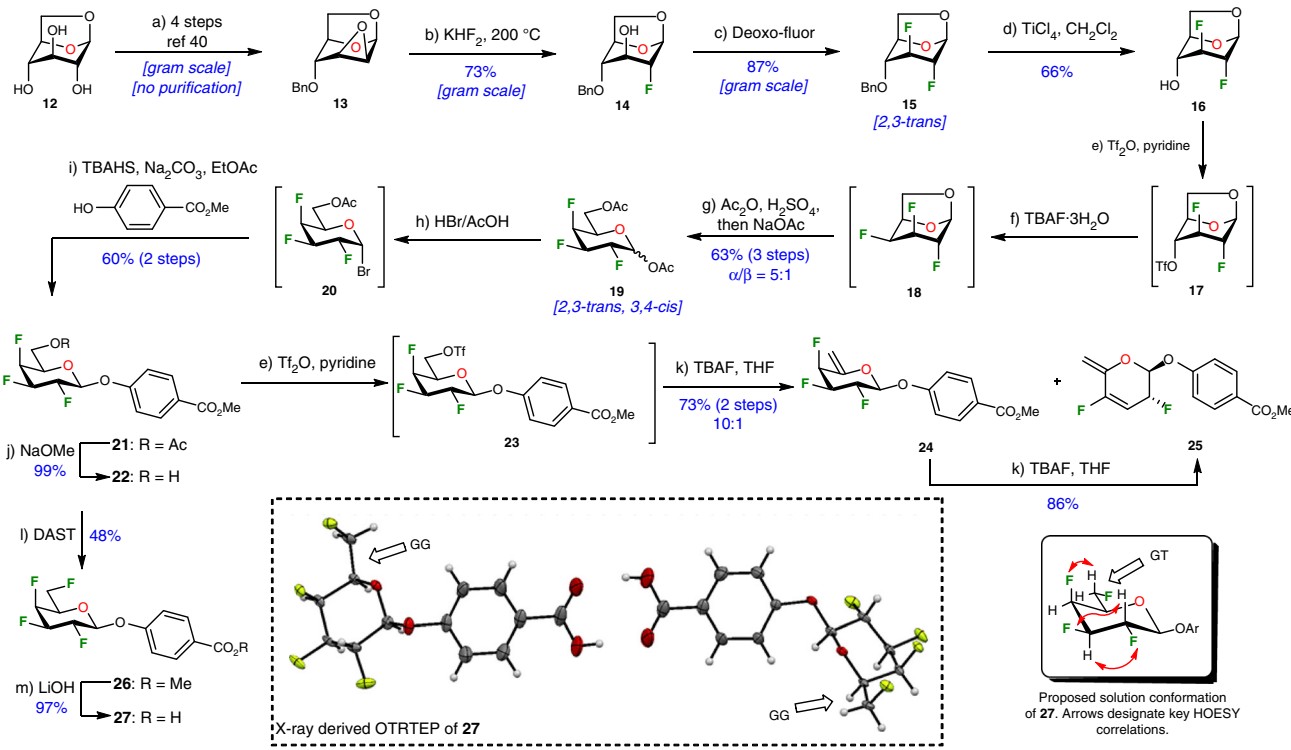

**Fig. 2** Stereoselective synthesis of 2,3,4,6-tetradeoxy-2,3,4,6-tetrafluorogalactopyranoside **27** from levoglucosan **12**. Reagents and conditions: (a) ref [40]; (b) KHF$_2$ (7.0 equiv), ethylene glycol, 200 °C, 2.5 h, 73%; (c) Deoxo-Fluor (2.0 equiv), THF, microwave irradiation, 100 °C, 1.5 h, 87%; (d) TiCl$_4$ (1.1 equiv), CH$_2$Cl$_2$, 0 °C, 0.5 h, 66%; (e) Tf$_2$O (2.0 equiv), pyridine (3.0 equiv), 0 °C, 0.2 h; (f) TBAF·3H$_2$O (1.5 equiv), CH$_2$Cl$_2$, rt, 15 h; (g) Ac$_2$O (30 equiv), H$_2$SO$_4$ (10 equiv), 0 °C to rt, 18 h, then NaOAc (20 equiv), rt, 0.3 h, 63% over 3 steps, $\alpha/\beta$ = 5:1; (h) 33% HBr in AcOH, CH$_2$Cl$_2$, rt, 66 h; (i) methyl *p*-hydroxybenzoate (3.0 equiv), TBAHS (1.0 equiv), EtOAc, 1 M Na$_2$CO$_3$, rt, 18 h; 60% over 2 steps; (j) 1 M NaOMe, MeOH, rt, 1 h, 99%; (k) 1 M TBAF in THF (18 equiv), rt, 1 h (for **24**), 3 days (for **25**), 73% over 2 steps for **24**, 86% for **25**; (l) DAST (3.0 equiv), 2,4,6-collidine (6.0 equiv), CH$_2$Cl$_2$, microwave irradiation, 100 °C, 1 h, 96%, **26/24** = 1:1; (m) 1 M LiOH (3.5 equiv), H$_2$O/MeOH/THF (2:3:5), 97%. *Ac$_2$O* acetic anhydride, *DAST* diethylaminosulfur trifluoride, *Deoxo-Fluor bis*(2-methoxyethyl)aminosulfur trifluoride, *TBAF* tetrabutylammonium fluoride, *TBAHS* tetrabutylammonium hydrogen sulfate, *Tf$_2$O* trifluoromethanesulfonic anhydride. ORTEP diagram of the molecular structure of **27** showing 50% thermal ellipsoid probability, carbon (gray), oxygen (red), fluorine (light green), hydrogen (white)

Calculations were performed with the CAM-B3LYP functional[56] using Grimme's D3 correction[57] and the 6–31 + G(d,p) basis set. The polarizable continuum model (PCM) was used to study possible solvent effects in acetone (for details, see Supplementary Discussion).[58,59] The molecular dipole moment of the three staggered conformations corresponding to the rotation about the C5-C6 bond were computed. That of the GG conformer is 5.29 D, in strong contrast with the other two conformers (GT conformer: 3.49 D and TG conformer: 2.03 D). Following this observation, we performed a Natural Bonding Orbital (NBO) analysis[60,61] to evaluate the possibility of hydrogen bonding involving fluorine atoms. Specifically, we looked at the NBO populations of the lone pairs on the fluorine centers (donors), as well as the C–H anti-bonding orbitals (acceptors) and the results are presented in the supplementary information. Briefly, no appreciable portion of the lone pair NBO population is donated to the antibonding pairs. This suggests that intermolecular C–F···H–C interactions are very weak. However, to the best of our knowledge, this is the first example of facially polarized organofluorine possibly responsible for crystal packing at the expense of strong 1,3-C–F bond repulsion. Finally, an HOESY (Heteronuclear Overhauser Effect Spectroscopy) experiment was performed on compound **27** (Acetone-$d_6$) in order to determine the conformation of the fluoromethyl group (C5–C6 linkage). Key correlations are presented in Fig. 2 and suggested that the tetrafluorinated galactoside derivative prefers a GT conformation in solution (Acetone-$d_6$).[48]

This was also confirmed after analysis of the $^1$H NMR spectrum (500 MHz, Acetone-$d_6$). The proton at C-5 has a chemical shift of 4.52 ppm with, amongst others, a coupling constant $^3J_{H5\text{-}F6}$ of 12.9 Hz, corresponding to a gauche conformation with F-6 (for details, see Supplementary Fig. 52). The results of our modeling calculations also support this finding. The rotation about the C5–C6 bond was scanned and results indicated three minima corresponding to the staggered conformations (Fig. 3b). The GG conformer, corresponding to the one from the crystal structure, is the least favorable ($\Delta G$ = 1.41 kcal/mol) as compared with the GT conformer ($\Delta G$ = 0.00 kcal/mol) and the TG conformer ($\Delta G$ = 0.20 kcal/mol). The calculated free energy difference is small and suggests that both GT and TG conformers will be present at room temperature.

**Synthesis of 2,3,4-trifluorinated hexopyranoses.** Encouraged by the successful synthesis of the first tetrafluorinated carbohydrate (where all the hydroxyl groups were replaced by fluorine atoms), we maintained our efforts to perform standard synthetic derivatizations from key-synthon **22** in order to prepare phospho-galactopyranoside, galacturonic acid methyl ester, and fucoside derivative (Fig. 4). Thus, starting from hydroxyl **22**, direct phosphorylation afforded phospho-galactopyranoside **28** in 84% yield. In addition, a TEMPO/BAIB-mediated oxidation allowed the formation of the corresponding uronic acid, directly treated in situ with methyl iodide under basic conditions to provide the

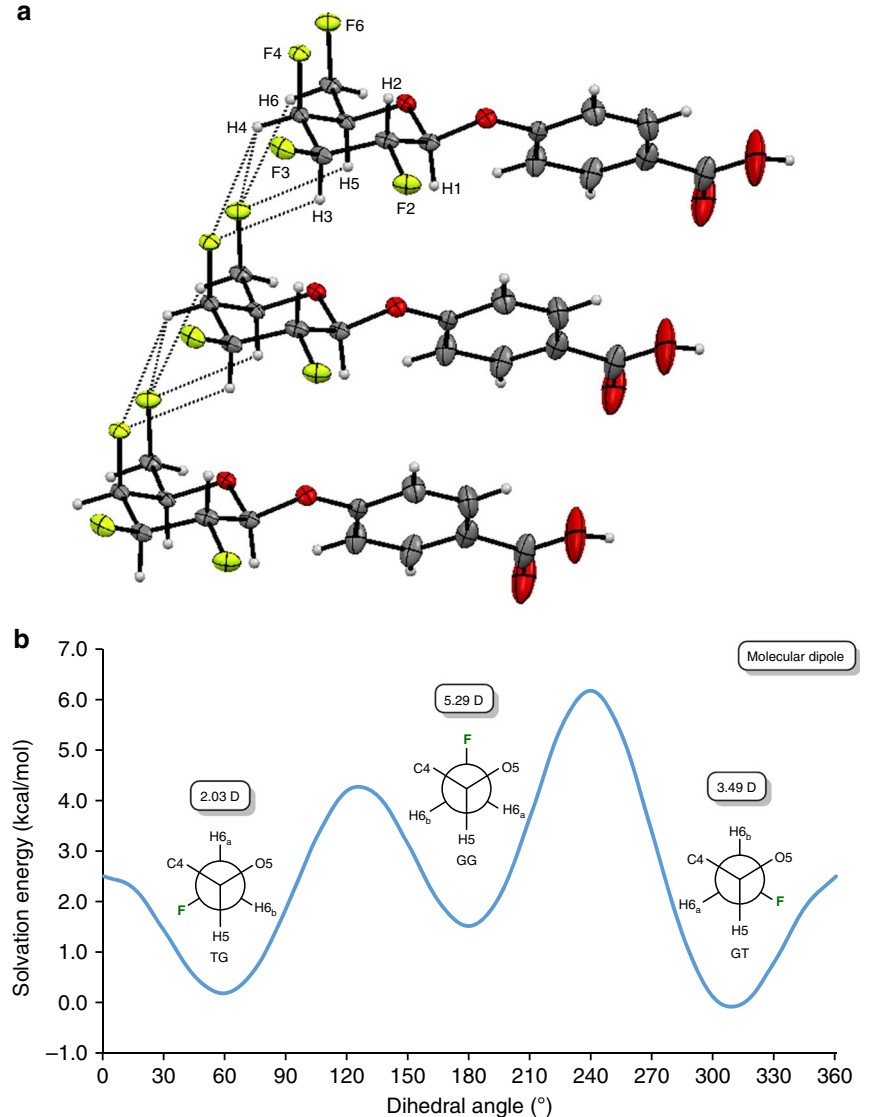

**Fig. 3** Conformation of compound **27**. **a** Crystal molecular packing arrangement of galactoside analog **27** highlighting possible C−F⋯H−C interactions (dot line). ORTEP diagram showing 50% thermal ellipsoid probability, carbon (gray), oxygen (red), fluorine (light green), hydrogen (white); **b** Relative energy of the dihedral angle plot (CAM-B3LYP-D3/6-31 + G(d,p)) for rotation about C5−C6 bond for compound **27**, along with their calculated molecular dipole moment (shown in the frame)

galacturonic acid methyl ester analog **29**. Finally, compound **22** was further deoxygenated at C-6 starting with installation of iodine, followed by a radical deiodination allowing the clean isolation of D-fucopyranoside **30** in a satisfactory yield. These results indicated that even if compound **22** is prone to elimination (see Fig. 2), standard modifications at C-6 can be accomplished on this complex fluorinated carbohydrate with only minor modifications of existing synthetic techniques. As such, these synthetic developments could be initiators of complex derivatizations and anticipated as straightforward methodologies towards the incorporation of these heavily fluorinated carbohydrate analogs into biologically active molecules.

We next turned our attention to the synthesis of contiguous stereogenic center with other *cis, trans* chemical relationships between fluorine atoms at position C-2, C-3, and C-4. The galactose derivative **19** integrates a *2,3-trans, 3,4-cis* relationship and the preparation of a fluorinated analog with a *2,3-trans, 3,4-trans* relationship could be straightforward from intermediate **16**

(Fig. 2). The preparation of the 2,3,4-trideoxy−2,3,4-trifluoroglucopyranose analog **33** is summarized in Fig. 5. Intermediate **16** was subjected to a Lattrell-Dax epimerization via triflate **17** allowing the quasi-quantitative formation of the 1,6-anhydrogalactopyranose derivative **31**. Nucleophilic fluorination at C-4 was achieved using TBAF via a triflate derivative and subsequent acetolysis furnished the acetyl protected *2,3-trans, 3,4-trans* 2,3,4-trideoxy-2,3,4-trifluoroglucopyranose **8** ([19]F NMR (470 MHz, CDCl$_3$) $^3J_{F2-H3}$ = 13.0 Hz, $^3J_{F4-H3}$ = 16.1 Hz, $^3J_{F4-H5}$ = 4.2 Hz, for details, see Supplementary Fig. 81) from the late acetylation of the intermediate **32**.[42] Standard deprotection under basic conditions furnished known glucose derivative **33**, initially prepared by the group of O'Hagan in 15 steps (0.4% global yield) from butynediol.[62] In comparison, the present strategy only required a 9-step sequence (only 6 purifications) from Cerny's epoxide **13** with an overall 25% yield. Of interest, this Chiron approach avoided the formation of cumbersome enantiomers' mixtures commonly encountered in de novo synthetic approaches.

As a demonstration of the usefulness of levoglucosan as starting material for the construction of fluorinated carbohydrates, we successfully completed the synthesis of 2,3,4-trideoxy-2,3,4-trifluoromannopyranose **44** and 2,3,4-trideoxy-2,3,4-trifluorotalopyranose **47** as shown in Fig. 6. The first step transformed the 1,6-anhydroglucose into intermediate **34** on a multigram-scale via a described five-step protocol necessitating only one purification.[63] Reaction of **34** with KHF$_2$ directly

furnished the desired 3-deoxy-3-fluoroglucopyranose **35** in a 65% yield. Subsequent fluorination of the C-2 position was achieved upon exposure to TBAF on triflate intermediate **36**. As a result, 2,3-dideoxy-difluoromannose derivative **37**, which possessed the necessary 2,3-*cis* relationship, was isolated in 85% yield ([19]F NMR (470 MHz, CDCl$_3$) $^3J_{F3-F2}$ = 4.6 Hz, for details, see Supplementary Fig. 92). The next step engaged intermediate **37** in a TiCl$_4$-mediated benzyl deprotection and generated intermediate **38**, which represented the perfect candidate for a nucleophilic fluorination reaction. Deoxofluorination of the axial C-4 hydroxyl group failed using standard methods or using a triflate as an activating group. At this point, we were compelled to invert the C-4 stereocenter in order to achieve successful fluorination. Thus, epimerization of the C-4 hydroxyl group **38** gave pivotal building-block **40** via triflate **39** in excellent yields. Once compound **40** was activated as triflate **41** and treated with Et$_3$N·3HF, 2,3,4-trifluoromannopyranose **42** was obtained as the major product (together with its C-4 epimer, not shown). Acetolysis of the crude reaction mixture furnished the desired 2,3-*cis*, 3,4-*trans* product **9** and its chromatographically separable C-4 diastereoisomer. In sharp contrast, deoxofluorination of **40** using DAST directly furnished exclusively the 2,3-*cis*, 3,4-*cis* product **10** ([19]F NMR (470 MHz, CDCl$_3$) $^3J_{F2-H3}$ = 29.2 Hz, $^3J_{F4-H3}$ = $^3J_{F4-H5}$ = 28.1 Hz, for details, see Supplementary Fig. 123), after facile acetolysis of intermediate 1,6-anhydro-2,3,4-trifluorotalopyranose **45**. The configuration of **9** and **10** were unambiguously confirmed by X-ray crystallographic analysis (see ORTEP, Fig. 6) through suitable derivatization of the corresponding diols **43** and **46**, generating the *p*-bromobenzoate derivatives **44** and **47**, respectively (benzoate moieties have been omitted for clarity).

With the continuous objective of developing a straightforward synthesis of trifluorinated hexopyranose, we opted to start with *bis*-tosylate **48**, readily accessible from levoglucosan in a multi-gram scale (Fig. 7).[64] Upon extensive experimentation, we discovered that treatment of **48** with KHF$_2$ (4 equiv.) and TBAF·3H$_2$O (8 equiv.) neat at 180 °C for 24 h led to the formation of 1,6-anhydro-2,4-dideoxy-2,4-difluoroglucopyranose **49** in 60% yield.[65] This procedure successfully allowed the stereoselective incorporation of two fluorine atoms placed 1,3-*syn* on a pyranose ring. The formation of intermediates **50**−**52** may be speculated to rationalize this transformation.[66] In this context, a series of epoxide formation-opening sequence with nucleophilic

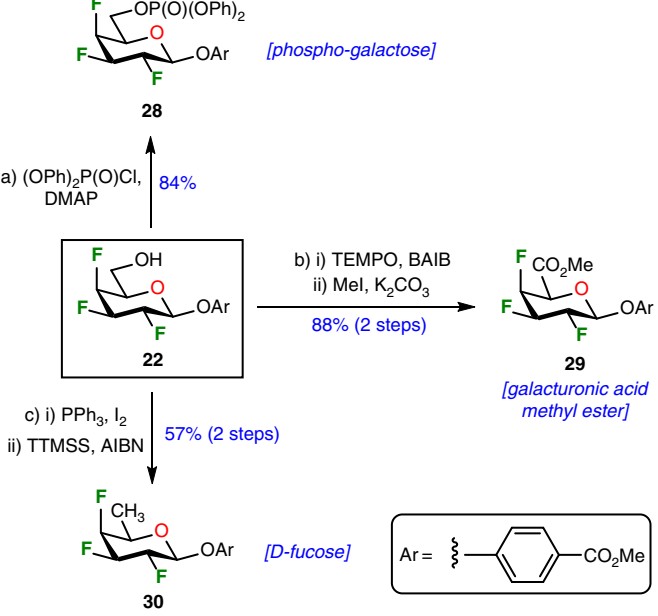

**Fig. 4** Derivatization of 2,3,4-trifluorinated galactopyranoside **22**. Reagents and conditions: (a) ClP(O)(OPh)$_2$ (1.5 equiv.), DMAP (1.5 equiv.), CH$_2$Cl$_2$, rt, 16 h, 84%; (b) (i)TEMPO (0.2 equiv.), BAIB (2.5 equiv.), CH$_2$Cl$_2$/H$_2$O, rt, 1.0 h, (ii) MeI (40 equiv.), K$_2$CO$_3$ (1.1 equiv.), CH$_3$CN, rt, 18 h, 88% over 2 steps; (c) (i) PPh$_3$ (1.5 equiv.), I$_2$ (1.5 equiv.), imidazole (2.0 equiv.), THF, reflux, 2.5 h, (ii) TTMSS (2.0 equiv.), AIBN (0.1 equiv.), toluene, reflux, 18 h, 56% over 2 steps. *AIBN* 2,2'-azobis(2-methylpropionitrile), *BAIB* bis(acetoxyiodo)benzene, *CH$_3$CN* acetonitrile, *DMAP* 4-(dimethylamino)pyridine, *PPh$_3$* triphenylphosphine, *TEMPO* 2,2,6,6-tetramethyl-1-piperidinyloxy, *THF* tetrahydrofuran, *TTMSS* tris(trimethylsilyl)silane

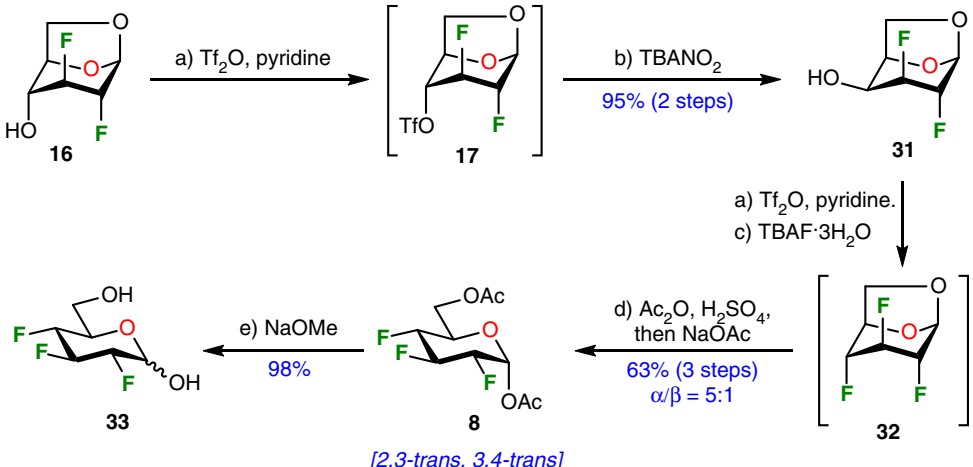

**Fig. 5** Stereoselective synthesis of 2,3,4-trideoxy-2,3,4-trifluoroglucopyranose **33**. Reagents and conditions: (a) Tf$_2$O (2.0 equiv), pyridine (3.0 equiv), CH$_2$Cl$_2$, 0 °C, 0.5 h; (b) TBANO$_2$ (3.0 equiv), CH$_3$CN, microwave irradiation, 100 °C, 3 h, 95% over 2 steps; (c) TBAF·3H$_2$O (1.5 equiv), CH$_2$Cl$_2$, rt, 18 h; (d) Ac$_2$O (30 equiv), H$_2$SO$_4$ (10 equiv), 0 °C to rt, 18 h, then NaOAc (20 equiv), rt, 0.3 h, 63% over 3 steps, α/β = 5:1; (e) 1 M NaOMe, MeOH, rt, 1 h, 98%. *Ac$_2$O* acetic anhydride, *TBAF* tetrabutylammonium fluoride, *TBANO$_2$* tetrabutylammonium nitrite, *Tf$_2$O* trifluoromethanesulfonic anhydride

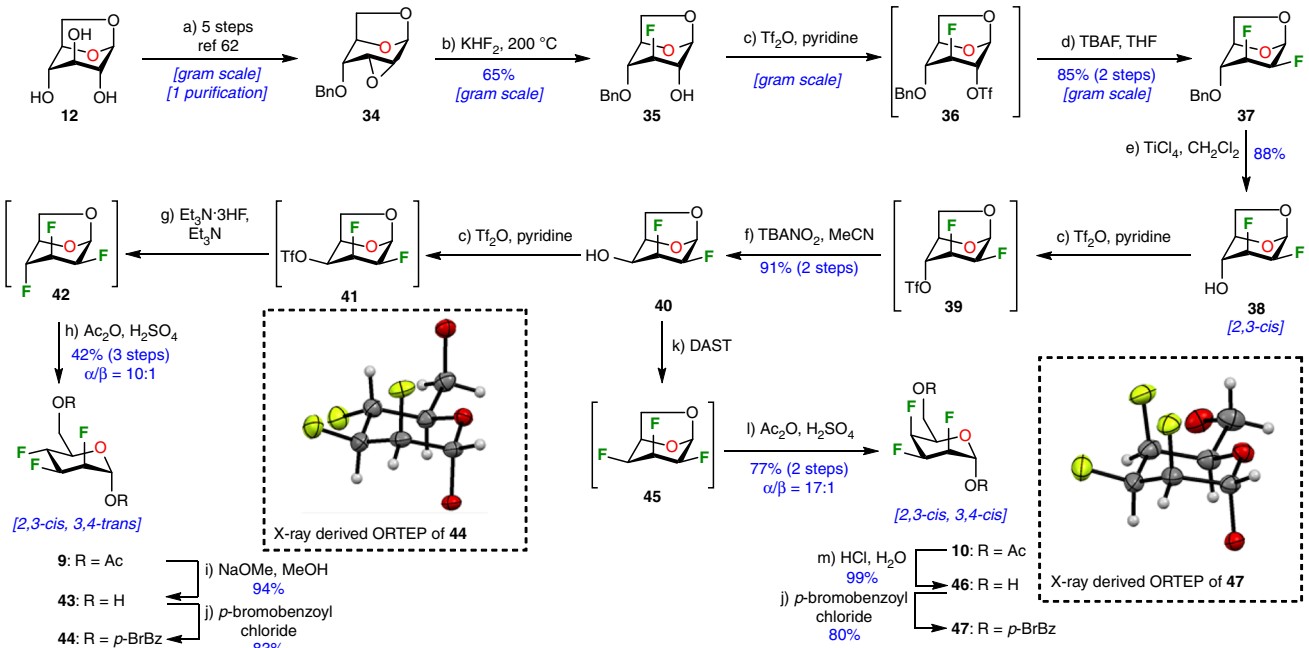

**Fig. 6** Synthesis of 2,3,4-trideoxy-2,3,4-trifluoromannopyranose **44** and 2,3,4-trideoxy-2,3,4-trifluorotalopyranose **47**. Reagents and conditions: (a) ref[63]; (b) KHF$_2$ (6.1 equiv), ethylene glycol, 200 °C, 5.0 h, 65%; (c) Tf$_2$O (2.4 equiv), pyridine (9.6 equiv), CH$_2$Cl$_2$, 0 °C, 0.5 h; (d) 1 M TBAF in THF (10 equiv), THF, rt, 22 h, 85% over 2 steps; (e) TiCl$_4$ (2.0 equiv), CH$_2$Cl$_2$, 0 °C, 0.8 h, 88%; (f) TBANO$_2$ (3.0 equiv), CH$_3$CN, microwave irradiation, 100 °C, 3 h, 91% over 2 steps; (g) Et$_3$N·3HF (15 equiv), Et$_3$N, 80 °C, 48 h; (h) Ac$_2$O (200 equiv), H$_2$SO$_4$ (80 equiv), 0 °C to rt, 16 h, then NaOAc (100 equiv), 0 °C to rt, 0.3 h, 71% over 3 steps, **9/10** = 1.5:1, α/β = 10:1; (i) 1 M NaOMe, MeOH, rt, 1 h, 94%; (j) p-bromobenzoylchloride (4.0 equiv), Et$_3$N (8 equiv), DMAP (0.8 equiv), CH$_2$Cl$_2$, rt, 18 h, 83% for **44**, 80% for **47**; (k) DAST (2.0 equiv), CH$_2$Cl$_2$, microwave irradiation, 100 °C, 1 h; (l) Ac$_2$O (30 equiv), H$_2$SO$_4$ (10 equiv), 0 °C to rt, 16 h, then NaOAc (20 equiv), 0 °C to rt, 0.3 h, 77% over 3 steps, α/β = 23:1; m) HCl (37% in water), water, rt, 1 h. Ac$_2$O acetic anhydride, DAST diethylaminosulfur trifluoride, DMAP 4-(dimethylamino)pyridine, TBAF tetrabutylammonium fluoride, TBANO$_2$ tetrabutylammonium nitrite, Tf$_2$O trifluoromethanesulfonic anhydride. ORTEP diagram of the molecular structure of **44** and **47** showing 50% thermal ellipsoid probability, carbon (gray), oxygen (red), fluorine (light green), hydrogen (white). The ORTEP plots do not show the complete molecule (benzoate moieties have been omitted for clarity)

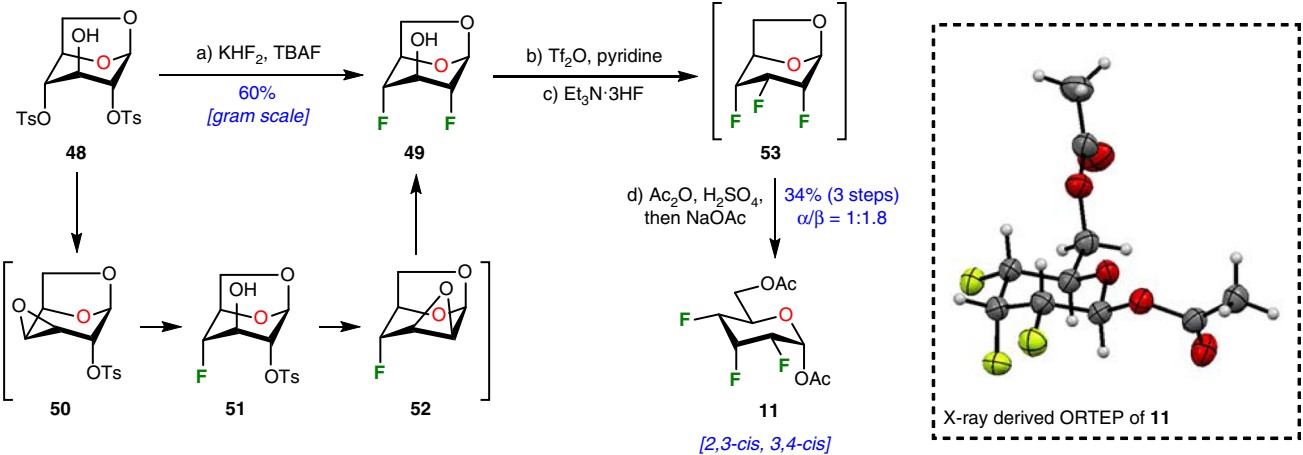

**Fig. 7** Rapid synthesis of 2,3,4-trideoxy-2,3,4-trifluoroallopyranose **11**. Reagents and conditions: (a) KHF$_2$ (4 equiv), TBAF·3H$_2$0 (8 equiv), 180 °C, 24 h, 60%; (b) Tf$_2$O (1.5 equiv), pyridine (3 equiv), CH$_2$Cl$_2$, rt, 0.5 h; (c) Et$_3$N·3HF (100 equiv), 120 °C, 20 h; (d) Ac$_2$O (30 equiv), H$_2$SO$_4$ (10 equiv), 0 °C to rt, 16 h, then NaOAc (20 equiv), rt, 0.3 h, 34% over 3 steps, α/β = 1:1.7. Ac$_2$Oacetic anhydride, NaOAc sodium acetate, TBAF tetrabutylammonium fluoride, Tf$_2$O trifluoromethanesulfonic anhydride. ORTEP diagram of the molecular structure of **11** showing 50% thermal ellipsoid probability, carbon (gray), oxygen (red), fluorine (light green), hydrogen (white)

fluorine would be distributed over a one-pot 4-step process (~88% yield per step), involving the break of 2 C−O bonds and the concomitant formation of 2 C−F bonds. Of interest, the application of the procedure at a multigram-scale reaction allowed the isolation of about 5 grams of *bis*-fluorinated

carbohydrate analog **49** in one single batch. At this point, preparation of the 2,3,4-trideoxy-2,3,4-trifluoroallopyranose followed a standard procedure. Thus, intermediate **49** was activated as triflate and treated with Et$_3$N·3HF allowing the formation of only one diastereoisomer **53** corresponding to the inversion of

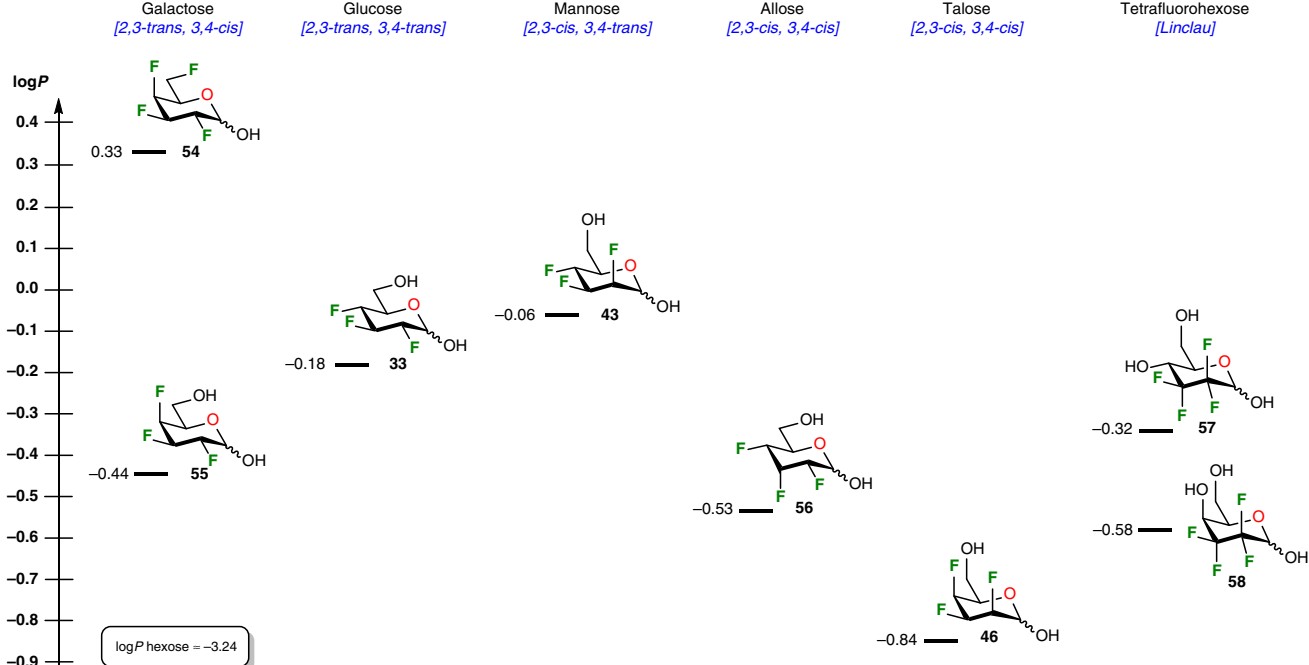

**Fig. 8** Newman projection of fluorinated pyrans. Talopyranose derivative **47** owns 1,3-diaxial repulsion between C−F bonds

**Fig. 9** Lipophilicities of fluorinated carbohydrates. Compounds **33**, **43**, **46**, **54**−**56** were prepared in this study and tetrafluorohexose analogs **57** and **58** were synthesized by the group of Linclau

configuration at C-3. Finally, acetolysis yielded fluorinated allopyranose derivative **11** (2,3-*cis*, 3,4-*cis* product) in 34% over 3 steps, requiring only 2 chromatographic purifications starting from *bis*-tosylate **48**. The stereochemistry of the fluorine atoms was unambiguously proven by $^{19}$F NMR ((470 MHz, CDCl$_3$) $^3J_{F3\text{-}H2}$ = $^3J_{F3\text{-}H4}$ = 25.6 Hz, $^3J_{F4\text{-}H5}$ = 1.9 Hz, for details, see Supplementary Fig. 145) and X-ray crystallographic analysis of compound **11**. This landmark synthesis shows that rapid access to complex fluorinated organic molecule is possible and continuous research should be directed towards this goal.

Interesting features can be addressed upon closer look at the torsion angle of the $^4C_1$ chair conformations for compounds **27**, **44**, **47**, and **11** in the solid state (Fig. 8). In order to avoid *syn*-1,3-difluoro contact, talose derivative **47** shows significant intra-annular torsion angle.[67] This repulsion leads to a greater distance between fluorine atoms at C-2 and C-4 of the pyran ring (2.817 (2) Å). Based on this observation, it can be concluded that 2,3,4-trideoxy-2,3,4-trifluorotalopyranose **47** preserves a slightly different shape and conformation as compared to other trifluorinated hexopyranoses and as compared with α-D-talopyranose in the solid state (distance O2−O4 = 2.655(4) Å).[68] As for the compounds **27**, **44**, and **11** they adopt standard $^4C_1$-like conformation and this was also confirmed using NMR analysis (for details, see Supplementary Tables).

**Lipophilicities of fluorinated carbohydrates.** Fluorination can be effective for lipophilicity (log$P$) modulation, as recently demonstrated by the group of Linclau that ascertained the lipophilicity of two tetrafluorinated hexoses (compound **57** and **58**, Fig. 9).[69] Interestingly, their rational investigation drawn preliminary trends correlating fluorination and lipophilicity for fluorinated carbohydrates. More particularly, large lipophilicity variations were observed within a same family of fluorinated carbohydrates in which hydroxyl group at C-4 had different relative configurations or replaced with a fluorine substituents.[69] In order to complement the investigation and to evaluate the influence of several structural parameters including the position, the stereochemistry, and the number of integrated fluorination sites in our set of fluorinated compounds, we used a log$P$ determination method developed recently by the same group, based on $^{19}$F NMR spectroscopy (for details, see Supplementary Figs. 166−175). First of all, among all trifluorinated analogs, both all-*cis* analogs (**46** and **56**) were the most hydrophilic. This is probably due to the increase in the overall dipole moment cause by facially polarized C−F bonds. Nevertheless, a significant difference of about 0.3 log$P$ unit was observable, highlighting that the inversion of the facial polarization contribute to a marked lipophilicity differentiation. In comparison, all-*trans* glucose derivative **33** is more lipophilic with a log$P$ value of −0.18. Epimerization at C-2 (mannose derivative **43**) is responsible for an

increase in lipophilicity (log$P$ value of −0.06), while epimerization at C-4 (galactose derivative **55**) resulted in an important decrease in the log$P$ value (−0.44 corresponding to $\Delta$log$P = 0.26$). This trends is also emphasized by the large log$P$ difference ($\Delta$log$P = 0.78$) between diastereoisomers **46** (*2,3-cis, 3,4-cis*) and **43** (*2,3-cis, 3,4-trans*), which only differs in the stereochemistry of the C-4 fluorine atom. These observations thus reinforced previous trends and suggested that both stereochemistry and substitutions on C-4 contribute to the pivotal role of this position regarding the overall lipophilicity. Finally, as expected, tetrafluorinated galactose analog **54** was more lipophilic with a log$P$ value of 0.33.

## Discussion

The synthesis of organofluorine compounds with the aim of discovering new chemical entities possessing unique and unsuspected physical, chemical and biological properties have attracted attention over the past years. In order to make our own contribution en route to this goal, the stereoselective synthesis of a family of polyfluorinated hexopyranoses was accomplished using a Chiron approach. Structural analysis of the original 2,3,4,6-tetradeoxy-2,3,4,6-tetrafluorohexopyranoside analog of galactose indicated that crystal packing overcompensates the 1,3-C−F bond repulsion. This was corroborated with DFT calculations of the relative energy and molecular dipole moments of the three staggered conformations corresponding to the rotation about the C5-C6 bond. The flexibility of the developed strategy led to the preparation of 2,3,4-trideoxy-2,3,4-trifluoro glucose, mannose, talose, fucose, and galacturonic acid methyl ester. Also, the rapid access to 2,3,4-trideoxy-2,3,4-trifluoroallopyranose was possible via a one-step operation, 4-step process that allow to construct 2 C−F bonds in high yield and stereoselectivity. All the fluorinated hexopyranoses were found to conserve their $^4C_1$ conformation. However, a slightly distorted conformation due to repulsive 1,3-diaxial F⋯F interaction was observed in the talose analog. Also, the lipophilicities of fluorinated carbohydrates were measured and it was notably determined that the relative stereochemistry of multi-vicinal fluorine atoms had a strong effect on the log$P$ value. By blending organic synthesis, method development endeavors and carbohydrate chemistry, we strongly believe that the resulting molecules and the associated synthetic protocols could serve as useful tools to deepen investigations on the use of intriguing fluorine-containing carbohydrate analogs and to underscore their relevance and their yet underestimated potential to chemistry, biology, and material sciences.

## Methods

**General information**. Unless otherwise stated, all reactions were carried out under an argon atmosphere with dry solvents and under anhydrous conditions. Tetrahydrofuran (THF) was distilled from sodium/benzophenone and dichloromethane ($CH_2Cl_2$) was distilled from calcium hydride immediately before use. Reactions were monitored by 0.20 μm Silicycle silica gel plates (F-254) thin-layer chromatography (TLC) using UV light as visualizing agent and using a TLC stain (solution of 3 g of PhOH and 5 mL of $H_2SO_4$ in 100 mL of EtOH). Flash column chromatography were performed using SiliaFlash® P60 40–63 μm (230–400 mesh). Nuclear magnetic resonance (NMR) spectra were recorded with an Agilent DD2 500 MHz spectrometer and calibrated using residual undeuterated solvent (CDCl₃: $^1$H $\delta = 7.26$ ppm, $^{13}$C $\delta = 77.16$ ppm; acetone-$d_6$: $^1$H $\delta = 2.05$ ppm, $^{13}$C $\delta = 29.8$ ppm) as an internal reference. Calibration of $^{19}$F NMR was performed using hexafluorobenzene, which have been measured at −162.29 ppm compared to the chemical shift of reference compound CFCl₃. Coupling constants ($J$) followed these abbreviations to designate multiplicities: br: broad, s: singlet, d: doublet, t: triplet, q: quartet, p: quintet, m: multiplet (reported in Hertz (Hz)). Assignments of NMR signals were made by homonuclear (COSY) and heteronuclear (HSQC, HMBC, HOESY, $^{19}$F c2HSQC) two-dimensional correlation spectroscopy. A Thermo Scientific Nicolet 380 FT-IR spectrometer was used to record infrared spectra (absorptions are given in wavenumbers: cm$^{-1}$). An Agilent 6210 LC Time of Flight mass spectrometer was used to collect high-resolution mass spectra (HRMS). Electrospray mode in either protonated molecular ions $[M + nH]^{n+}$, ammonium adducts $[M + NH_4]^+$ or sodium adducts $[M + Na]^+$ were used for confirmation of

the empirical formula. A JASCO DIP-360 digital polarimeter was used to record optical rotations (reported in units of $10^{-1}$ (degree cm$^2$ g$^{-1}$)).

**Further experimental data**. For detailed experimental procedures, see Supplementary Methods, for crystallographic data of compounds **11**, **27**, **44**, and **47**, see Supplementary Tables 1, 7, 13, and 19. For detailed on theoretical calculations, see Supplementary Discussion. For NMR spectra of the synthesized compounds, see Supplementary Figures.

## Data availability

Crystallographic data are deposited at the Cambridge Crystallographic Data Centre (CCDC). CCDC 1848261, 1824901, 1824902, and 1837072 contain the supplementary crystallographic data for this paper. The data can be obtained free of charge from The Cambridge Crystallographic Data Centre via www.ccdc.cam.ac.uk/structures. All the data supporting the findings of this study are available as Supplementary Information, or from the corresponding author on request.

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

## Acknowledgements

This work was supported by the Natural Sciences and Engineering Research Council of Canada (NSERC), the Fonds de Recherche du Québec–Nature et Technologies and the Université Laval. D.L. thanks the Fonds de Recherche du Québec–Nature et Technologies for a postgraduate fellowship. We thank Pierre Audet for NMR spectroscopic assistance (Université Laval) and Thierry Marris for crystallographic assistance (Université de Montréal). This research was enabled in part by Calcul Québec, Compute Ontario, SHARCNET, and Compute Canada. Finally, the authors would like to thank Dr. Jean-François Paquin, Dr. Gino A. DiLabio and Dr. Yoann M. Chabre for proof reading of this article and for useful discussions.

## Author contributions

V.D., D.L., and J.St.-G. performed and analyzed the experiments. V.D., D.L., J.St.-G., and D.G. designed the experiments. P.A.J. performed the DFT calculations. V.D., P.A.J., and D.G. prepared the manuscript.

## Additional information

**Competing interests:** The authors declare no competing interests.

