## [Peer Review File · Nature Communications]

Reviewer #1 (Remarks to the Author):

This paper describes in detail the synthesis of various polyfluorinated sugars using levoglucosan as a common starting material. Since it is extremely difficult to introduce a plurality of fluorine atoms into continuous asymmetric carbons, it is very interesting and significant that all sugars in this article can be synthesized in relatively good yields.

However, new synthetic reactions are not used in particular, and such sugar syntheses, in which fluorine atoms are incorporated into continuous asymmetric carbons, have already been reported by O'Hagan as a pioneer. In addition, O'Hagan and colleagues mention the detailed structural features in their own papers, so there is no particular finding on structural specificity.

It can not be said that it is enough to publish in this magazine, if some new original findings by the authors are not added. For example, it should be better to add the correlation between the position of fluorine atom, the stereochemistry, the introduction number and the structure, from the X-ray crystal structure analysis of all sugar derivatives.

Reviewer #2 (Remarks to the Author):

Crystallographic report - relatively minor modifications.

This manuscript describes a new stereoselective synthetic approach to fluorinating a number of hexapyranoses.

It reports 4 crystal structures.

The crystallographic analysis has been competently performed and the structures are all of publishable quality. However, there are a number of issues relating to the crystal structure reporting that are detailed below. The main points are incorrect deposition codes and inconsistent versions of the structure refinement of compound 27. If these are addressed the crystallography will be at a satisfactory level for publication.

1) Deposition codes.

The deposition codes for the structures were incomplete, the correct ones are: Structure 27: CCDC# 1824899, Structure 44: CCDC# 1824901, Structure 47: CCDC# 1824902, Structure 11: CCDC# 1837072

Please amend the data availability text to read as follows:

“CCDC 1824899, 1824901, 1824902 & 1837072 contain the supplementary crystallographic data for this paper. The data can be obtained free of charge from The Cambridge Crystallographic Data Centre via www.ccdc.cam.ac.uk/structures.”

- 2) Scheme 1. The resolution of the Ortep figure should be increased.
- 3) Quoted bond lengths, angles and interactions should have estimated standard deviations – this applies throughout the manuscript.
- 4) Figure 2 does not appear to show the complete hydrogen bonding possibilities. This figure should be redrawn to aid the reader in visualizing the potential interactions. It would also be helpful to label the atoms referred to in the text.
- 5) Scheme 4. The caption should mention that the Ortep plots do not show the complete molecule.
- 6) All Orteps should state the thermal ellipsoid probability.
- 7) The structure of 27 deposited with the CCDC and that included in the supplementary information are not the same. The unit cell parameters differ slightly indicating a different single crystal experiment (rather than just additional least squares cycles of the structure refinement). Assuming the authors have access to two different determinations of the same structure, they must be consistent in which one they report (replacing the deposited CIF if necessary). It is also extremely important that the atom numbering scheme is consistent between the structure and the text – at present this does not appear to be the case when the hydrogen bond parameters are discussed.

Reviewer #3 (Remarks to the Author):

Compounds containing multiple C-F stereogenic center exhibit intriguing properties, notable examples pertaining to this work is hexafluorocyclohexane and trifluorinated glucose synthesized by

O'Hagan and co-workers (NC, 2015, 7, 483; ACIE, 2012, 51, 10086; Chem. Commun., 2010, 46, 5434–5436). However, the synthesis of these molecules is usually very challenging due to competing reactions such as elimination and rearrangement.

The manuscript by Giguère and co-worker describes the authors' investigations into the stereoselective synthesis of carbohydrates with continuous C-F stereogenic center. The Chiron pool approach developed in this work has the advantage of great flexibility, which results in a collective synthesis of polyfluorinated carbohydrates, including those with D-glucose, D-mannose and D-talose configuration; many of these molecules (six in seven? the authors should make this clear) are unprecedented. Moreover, the syntheses are efficient in terms of overall yields, regioselectivity and stereoselectivity, allowing relatively easy access to a series of polyfluorinated carbohydrates.

Given the potential applications of polyfluorinated carbohydrate and the challenge to access them, the reported work will be of considerable interest to those in the synthesis and biochemistry communities. The research is well described and the supporting information is sufficient to demonstrate that the target products were indeed prepared. However, one main drawback is that the authors failed to demonstrate the application of these polyfluorinated carbohydrate, though the obtained molecules showed some unique properties (e.g. the alignment of C-F with 1,3-repulsion were observed in solid state). The authors should conduct some more experiments (e.g. calculation and/or biological testing) to strength this part. Thus, it is the opinion of this referee that the manuscript could be suitable for publication in Nature Communications after major revision.

1. The authors discussed the interesting molecular shape in solid state of compounds 27, 44, 47, and 11, but put the description of conformation analysis in the SI. I think it is more informative to use some words in the main text.
2. The CCDC number provided for compounds 27, 44, 47 are probably wrong, as all the number is 850759 (ref. 28, 35). The authors should be very careful when resubmitting the revised manuscript.
3. The one-pot, 4-step reaction (maybe one-step operation, 4-step reaction is better) for the converting of 48 to 49 is very impressive, given the fact the difficulty for the installation of multiple fluorine atoms stereoselectively. Are there any precedents for the in-situ generation of epoxide and opening with this reagent combination, or some similar reagents? Are there any reactions to introduce two or more fluorine atoms stereoselectively in one step operation fashion? If yes, it is better to make a comparison.
4. The alignment of 1,3-C-F bond in compound 27 is interesting, as usually this would result a strong repulsion effect. The authors proposed molecular dipole moment is responsible. I am wondering whether it is possible to attest this effect by calculation (within the publication timeframe.)
5. it is better to show the readers some biological or physical applications of these targets (an example: Chem. Commun., 2010, 46, 5434–5436); Also, I think it is better to combine the discussion of the solid state structures (figure 2 and figure 3) and put it at the end of results and discussion.

6. In the abstract, the authors state that "...towards the first stereoselective synthesis of several unprecedented polyfluorinated hexopyranoses", it is better to state how many, for example, six of which are unprecedented.
7. What is the correct abbreviation for j. fluorine chemistry, please check ref. 24 and 35.

All reviewers raised a few points that permitted lifting the quality of the manuscript. All comments are addressed in a point-by-point manner. Any changes from the original version of the manuscript are highlighted using a yellow background in the revised version.

Reviewers' comments:

Reviewer #1 (Remarks to the Author): *This paper describes in detail the synthesis of various polyfluorinated sugars using levoglucosan as a common starting material. Since it is extremely difficult to introduce a plurality of fluorine atoms into continuous asymmetric carbons, it is very interesting and significant that all sugars in this article can be synthesized in relatively good yields. However, new synthetic reactions are not used in particular, and such sugar syntheses, in which fluorine atoms are incorporated into continuous asymmetric carbons, have already been reported by O'Hagan as a pioneer. In addition, O'Hagan and colleagues mention the detailed structural features in their own papers, so there is no particular finding on structural specificity.*

It can not be said that it is enough to publish in this magazine, if some new original findings by the authors are not added. For example, it should be better to add the correlation between the position of fluorine atom, the stereochemistry, the introduction number and the structure, from the X-ray crystal structure analysis of all sugar derivatives.

Reviewer 1 suggested that “new original findings” should be added to this manuscript. In order to comply with these legitimate requirements, we took a couple of months to significantly improve the previous version of the manuscript. In that context, we decided to team up with a computational chemist to achieve a higher standard of understanding regarding the molecular ordering of the polyfluorinated galactoside **27** in the solid state. Using DFT, we found the relative energy of the three staggered conformations of this compound and we were able to determine the molecular dipole moment. Moreover, we looked at the NBO populations of the lone pairs for the GG conformer of compound **27**. These results suggested that intermolecular C–F...H–C interactions are not responsible for the molecular ordering. Beside theoretical calculations, we also evaluated the influence of the relative stereochemistry of fluorine atoms, over lipophilicity. The lipophilicities of fluorinated carbohydrates were determined and the relative stereochemistry of multi-vicinal fluorine atoms has a strong effect on the log*P* value (with trifluorinated talose being the less lipophilic and trifluorinated mannose being the most lipophilic).

Reviewer #2 (Remarks to the Author): *Crystallographic report - relatively minor modifications. This manuscript describes a new stereoselective synthetic approach to fluorinating a number of hexapyranoses. It reports 4 crystal structures. The crystallographic analysis has been competently performed and the structures are all of publishable quality. However, there are a number of issues relating to the crystal structure reporting that are detailed below. The main points are incorrect deposition*

codes and inconsistent versions of the structure refinement of compound **27**. If these are addressed the crystallography will be at a satisfactory level for publication.

1) *Deposition codes. The deposition codes for the structures were incomplete, the correct ones are: Structure **27**: CCDC# 1824899, Structure **44**: CCDC# 1824901, Structure **47**: CCDC# 1824902, Structure **11**: CCDC# 1837072. Please amend the data availability text to read as follows: "CCDC 1824899, 1824901, 1824902 & 1837072 contain the supplementary crystallographic data for this paper. The data can be obtained free of charge from The Cambridge Crystallographic Data Centre via www.ccdc.cam.ac.uk/structures."*

2) *Scheme 1. The resolution of the ORTEP figure should be increased.*

3) *Quoted bond lengths, angles and interactions should have estimated standard deviations – this applies throughout the manuscript.*

4) *Figure 2 does not appear to show the complete hydrogen bonding possibilities. This figure should be redrawn to aid the reader in visualizing the potential interactions. It would also be helpful to label the atoms referred to in the text.*

5) *Scheme 4. The caption should mention that the ORTEP plots do not show the complete molecule.*

6) *All ORTEPS should state the thermal ellipsoid probability.*

7) *The structure of **27** deposited with the CCDC and that included in the supplementary information are not the same. The unit cell parameters differ slightly indicating a different single crystal experiment (rather than just additional least squares cycles of the structure refinement). Assuming the authors have access to two different determinations of the same structure, they must be consistent in which one they report (replacing the deposited CIF if necessary). It is also extremely important that the atom numbering scheme is consistent between the structure and the text – at present this does not appear to be the case when the hydrogen bond parameters are discussed.*

Reviewer 2 rightfully raised points related to the incorrect deposition codes and inconsistent versions of the structure refinement of compound **27**. All corrections, including these both points, were carefully addressed (see below).

1) We used the correct deposition codes: Structure **27**: CCDC# 1848261, Structure **44**: CCDC# 1824901, Structure **47**: CCDC# 1824902, Structure **11**: CCDC# 1837072. Also, we amended the data availability text to read as follows: "CCDC 1848261, 1824901, 1824902 & 1837072 contain the supplementary crystallographic data for this paper. The data can be obtained free of charge from The Cambridge Crystallographic Data Centre via www.ccdc.cam.ac.uk/structures."

2) .The resolution of the Ortep figure was increased.

3) We used Mercury as program to process CIF files. Mercury provides estimated standard deviations (ESDs) for bond lengths and angles. In the manuscript, we added ESD for any bond lengths, angles and interactions for non-hydrogen atoms. Any measured values involving hydrogen will not have ESDs as they would not be relevant (Mercury places hydrogens at "fixed" geometric positions, for more information, see:

<https://www.ccdc.cam.ac.uk/support-and-resources/support/case/?caseid=aef817e2-a1d8-451c-86fe-16e752e44a2a>).

4) We increased the resolution of Figure 2a. We also added the label of atoms (we used the common nomenclature of carbohydrates). Figure 2b indicating the Newman projection of compound **27** also shows partly the label of atoms. Figure 2a focuses on the possible C-F...H-C interactions, especially for F4 and F6. Showing the complete hydrogen bonding possibilities would overburden Figure 2a.

5) In scheme 4, we added a sentence stated that the ORTEP plots do not show the complete molecule.

6) We added the thermal ellipsoid probability for all ORTEPs. We also added in the caption the color code for every atoms.

7) For structure **27**, we had access to different determinations of the same structure. To be consistent, we deposited the CIF file to the CCDC. We use this new deposition number in the manuscript (CCDC# 1848261).

Reviewer #3 (Remarks to the Author): *Compounds containing multiple C-F stereogenic center exhibit intriguing properties, notable examples pertaining to this work is hexafluorocyclohexane and trifluorinated glucose synthesized by O'Hagan and co-workers (NC, 2015, 7, 483; ACIE, 2012, 51, 10086; Chem. Commun., 2010, 46, 5434–5436). However, the synthesis of these molecules is usually very challenging due to competing reactions such as elimination and rearrangement. The manuscript by Giguère and co-worker describes the authors' investigations into the stereoselective synthesis of carbohydrates with continuous C-F stereogenic center. The Chiron pool approach developed in this work has the advantage of great flexibility, which results in a collective synthesis of polyfluorinated carbohydrates, including those with D-glucose, D-mannose and D-talose configuration; many of these molecules (six in seven? the authors should make this clear) are unprecedented. Moreover, the syntheses are efficient in terms of overall yields, regioselectivity and stereoselectivity, allowing relatively easy access to a series of polyfluorinated carbohydrates. Given the potential applications of polyfluorinated carbohydrate and the challenge to access them, the reported work will be of considerable interest to those in the synthesis and biochemistry communities. The research is well described and the supporting information is sufficient to demonstrate that the target products were indeed prepared. However, one main drawback is that the authors failed to demonstrate the application of these polyfluorinated carbohydrate, though the obtained molecules showed some unique properties (e.g. the alignment of C-F with 1,3-repulsion were observed in solid state). The authors should conduct some more experiments (e.g. calculation and/or biological testing) to strength this part. Thus, it is the opinion of this referee that the manuscript could be suitable for publication in Nature Communications after major revision.*

1) *The authors discussed the interesting molecular shape in solid state of compounds **27**, **44**, **47**, and **11**, but put the description of conformation analysis in the SI. I think it is more informative to use some words in the main text.*

2) The CCDC number provided for compounds **27**, **44**, **47** are probably wrong, as all the number is 850759(ref. 28, 35). The authors should be very careful when resubmitting the revised manuscript.

3) The one-pot, 4-step reaction (maybe one-step operation, 4-step reaction is better) for the converting of **48** to **49** is very impressive, given the fact the difficulty for the installation of multiple fluorine atoms stereoselectively. Are there any precedents for the in-situ generation of epoxide and opening with this reagent combination, or some similar reagents? Are there any reactions to introduce two or more fluorine atoms stereoselectively in one step operation fashion? If yes, it is better to make a comparison.

4) The alignment of 1,3-C-F bond in compound **27** is interesting, as usually this would result a strong repulsion effect. The authors proposed molecular dipole moment is responsible. I am wondering whether it is possible to attest this effect by calculation (within the publication timeframe).

5) It is better to show the readers some biological or physical applications of these targets (an example: Chem. Commun.,2010, 46, 5434–5436); Also, I think it is better to combine the discussion of the solid state structures (figure 2 and figure 3) and put it at the end of results and discussion.

6) In the abstract, the authors state that "...towards the first stereoselective synthesis of several unprecedented polyfluorinated hexopyranoses", it is better to state how many, for example, six of which are unprecedented.

7) What is the correct abbreviation for j. fluorine chemistry, please check ref. 24 and 35.

The major point raised by reviewer 3 is the lack of application of these polyfluorinated carbohydrate and more experiments should be conducted. In that regards, we added theoretical calculation experiments to deeply understand the alignment of C-F 1,3-repulsion of compound **27** in the solid state. Moreover, in order to unveil new properties of our unprecedented carbohydrate analogs, we determined the octanol-water partition coefficient (logP) for all trifluorinated and tetrafluorinated sugars. To that end, we prepared unprotected fluorinated carbohydrates. We disclosed these results in the Supporting Information, along with the NMR related to logP determination. We also validated the LogP method by comparison with known 2,3,4-trideoxy-2,3,4-trifluoroglucose **33**, which value compared well with the one originally obtained (-0.18 vs -0.17).

1) NMR analysis proved to be very instructive to determine the conformation of fluorinated carbohydrates. All these data are available in the SI. Moreover, the conformation in the solid state of fluorinated sugars are compared with the conformation of natural sugars in the solid state (Table S2-S5, S8-S11, and S14-S17). Only the conclusion of this large data collection are presented in the manuscript.

2) The CCDC number for compounds **27**, **44**, **47** has been fixed.

3) Regarding the one-step operation, 4-step reaction, there is precedent for epoxide opening using KHF_2 and $\text{TBAF}\cdot 3\text{H}_2\text{O}$ (ref. 43 in the manuscript). Nevertheless, there is

no precedent for epoxide generation and opening with these reagents combination. To the best of our knowledge, there is no report of any reactions to introduce two fluorine atoms stereoselectively (placed 1,3 on a hydrocarbon chain or in a ring) in a one-step fashion. We added one sentence to reinforce this new process and we also added one more reference to support the proposed mechanism.

4) As stated above, we use theoretical calculation to determine molecular dipole moment of the three staggered conformations of compound **27**. We also established the molecular dipole moment of small cluster of the GG conformer of compound **27** (see SI).

5) In the manuscript, we indicated to the reader novel physical applications of our fluorinated carbohydrates. As stated above, we determined the $\log P$ values of all trifluorinated carbohydrates. Major trends goes as follow:

- i) Both all-*cis* analogs were the most hydrophilic, due to the increase in the overall dipole moment caused by facially polarized C-F bonds;
- ii) In contrast, all-*trans* glucose derivative **33** is more lipophilic ($\log P$ value of -0.18);
- iii) The axially positioned C-4 fluorine atom (*cis* to the hydroxymethyl group) generates a strong decrease in lipophilicity. Moreover, the discussion of the solid state of compound **27** (Figure 2) is not combined with Figure 3. We rather extended our analysis of the solid state of compound **27** in relation with conformation of the latter compound and molecular dipole moment (addition of Figure 2b).

6) In the abstract, we stated that "...towards the first stereoselective synthesis of several unprecedented polyfluorinated hexopyranoses" and we changed it to "the stereoselective synthesis of polyfluorinated hexopyranoses, six of which are unprecedented."

7) The exact abbreviation for Journal of fluorine chemistry is J. Fluorine Chem. and we corrected ref 35, accordingly.

Finally, we took the liberty of changing the Graphical abstract. We replaced the protected carbohydrate analogs by the unprotected one. The authors hope that this revised manuscript fulfills all the requested criteria for *Nature Communications*.

Reviewer #2 (Remarks to the Author):

Crystallographic report (revised manuscript).

This manuscript describes a new stereoselective synthetic approach to fluorinating a number of hexapyranoses.

It reports 4 crystal structures.

The suggested amendments and corrections have been made to the crystallography which is now of at a satisfactory level.

Accept.

Reviewer #3 (Remarks to the Author):

I am generally satisfied with the revision provided by the authors. However, there are still some issues that need to be addressed before this paper could be accepted.

1. The abstract is way too long, especially after the authors adding the information of the calculation and log p. This problem also exists for the conclusion part. The authors should extract the essence of this work and rewrite the abstract and conclusion.

2. Some typos are found and listed below.

a) equiv. should be equiv

b) Delete "moiety" in "the benzoate aglycone was ultimately transformed into the corresponding carboxylic acid moiety 27 with the use of aqueous 1M LiOH solution"

c) "give raise to" should be "give rise to".

d) "24 was easily prone to" should be "24 was prone to..."

Reviewer #4 (Remarks to the Author):

I have assessed the new theoretical additions to this manuscript and, in particular, the DFT calculations.

In my opinion the calculations have been performed in a competent manner using a level of theory that is capable of making the predictions that are required, in this case point internal energies and dipole moments. My only reservation would be that the calculations are performed in vacuum and without any explicit solvent molecules (the results are compared to solution state NMR measurements and dehydrated crystals), and solvent could have significant effects on the dipole moment. The authors could: (1) include some reference to potential solvent effects, both in terms of the DFT calculations and the aqueous environments that would/would not be present in NMR and x-ray diffraction experiments, or (2) perform further QM calculations with some explicit water molecules (which may not be too onerous these days).

I also didn't quite understand why the x-ray experiments were deemed to be "very instructive." At one level, and in my opinion on reading the new theory section, the conformation thrown up by the x-ray experiments could be seen to be confusing or artifactual. The authors mentioned that the observed conformation is unexpected and concluded that it is due to "molecular ordering." It was not immediately obvious to me what "molecular ordering" meant in this context, since molecular ordering occurs in both the solid state and in solution; I assume that they are referring to crystal packing? The authors could rewrite this making more explicit reference to crystal packing or solid state as opposed to molecular ordering and also highlighting that the crystal structure conformation is likely to be a solid state artifact once seen in the context of NMR experiments and DFT calculations, which is what I concluded. Furthermore, now considering that this conformation may be due to crystal packing and considering that they have access to the unit cell do they

understand and can they explain why? This would also help convince the reader.

All reviewers raised a few points that permitted lifting the quality of the manuscript. All comments are addressed in a point-by-point manner. Any changes from the original version of the manuscript are highlighted using a yellow background in the revised version.

Reviewers' comments:

Reviewer #2 (Remarks to the Author): *Crystallographic report (revised manuscript). This manuscript describes a new stereoselective synthetic approach to fluorinating a number of hexapyranoses. It reports 4 crystal structures. The suggested amendments and corrections have been made to the crystallography which is now of at a satisfactory level. Accept.*

We are pleased that Reviewer #2 accepted the amendments and corrections to our revised manuscripts.

Reviewer #3 (Remarks to the Author): *I am generally satisfied with the revision provided by the authors. However, there are still some issues that need to be addressed before this paper could be accepted.*

1. *The abstract is way too long, especially after the authors adding the information of the calculation and log p. This problem also exists for the conclusion part. The authors should extract the essence of this work and rewrite the abstract and conclusion.*
2. *Some typos are found and listed below.*
 - a) *equiv. should be equiv*
 - b) *Delete "moiety" in "the benzoate aglycone was ultimately transformed into the corresponding carboxylic acid moiety 27 with the use of aqueous 1M LiOH solution"*
 - c) *"give raise to" should be "give rise to".*
 - d) *"24 was easily prone to" should be "24 was prone to..."*

We are pleased that Reviewer #3 is satisfied with the revised manuscript. We managed the small issues that needed to be addressed.

1. We shortened the abstract (less than 150 words) and the conclusion to only extract the essence of our work.
2. We corrected the typos listed by this reviewer:
 - a) equiv. was change by equiv
 - b) the word "moiety" was deleted from the sentence stated above
 - c) "give raise to" was change to "give rise to"
 - d) the word "easily" was deleted from the sentence stated above

Reviewer #4 (Remarks to the Author): *In my opinion the calculations have been performed in a competent manner using a level of theory that is capable of making the predictions that are required, in this case point internal energies and dipole moments.*

We thank the Reviewer #4 for the positive comments regarding the calculations.

My only reservation would be that the calculations are performed in vacuum and without any explicit solvent molecules (the results are compared to solution state NMR

measurements and dehydrated crystals), and solvent could have significant effects on the dipole moment.

We understand the Reviewer's reservations with respect to solvent effects, and address these concerns below.

The authors could: (1) include some reference to potential solvent effects, both in terms of the DFT calculations and the aqueous environments that would/would not be present in NMR and x-ray diffraction experiments, or (2) perform further QM calculations with some explicit water molecules (which may not be too onerous these days).

To be clear, the NMR experiments were carried out in acetone- d_6 as the solvent, not water (our polyfluorinated sugar is not soluble in water). We performed additional calculations in order to assess the impacts of solvent. The inclusion of explicit acetone molecules is somewhat onerous (much more so than the inclusion of explicit water molecules). As acetone is a weakly coordinating solvent with our fluorinated sugar, it is difficult to assess where and how many explicit solvent molecules to use, and to do so consistently for all three conformers. As such we computed the relative energy of the conformers using an implicit acetone solvent model – namely the polarizable continuum model (as implemented in Gaussian 09). The results are as follows (CAM-B3LYP-D3/6-31+G(d,p)):

Free conformers in gas phase:

Conformer	Energy (kcal/mol)	Enthalpy (kcal/mol)	Gibbs' Free Energy (kcal/mol)	Dipole Moment (D)
GG	4.44	4.33	4.42	3.6439
GT	0.70	0.64	0.70	2.3735
TG	0.00	0.00	0.00	1.2847

Conformers with implicit solvent:

Conformer	Energy (kcal/mol)	Enthalpy (kcal/mol)	Gibbs' Free Energy (kcal/mol)	Dipole Moment (D)
GG	1.62	1.54	1.41	5.2859
GT	0.00	0.00	0.00	3.4897
TG	0.28	0.36	0.20	2.0293

Here the energy values are relative to the most stable conformer. The implicit solvation model increases the dipole moment of each conformer though the overall trend does not change. The trend in energies however has changed. The implicit solvation predicts the GT conformer to be the most stable. Since this free energy difference is smaller than the generally accepted chemical accuracy (~1 kcal/mol) we can only conclude that the GT and TG conformers are very close in energy, and that both are likely seen in solution at room temperature. The GG conformer is also stabilized by the implicit solvation. A rough calculation of the Boltzmann populations suggest that the GG conformer might account for 5.1% of the population at thermal equilibrium at 298K. In a dilute solution on an NMR timescale, this is probably too small to observe.

This information is now included on pages 7 and 8 of the manuscript. We have additionally performed the dihedral scan (**Figure 2b**) with implicit solvation. All the

quoted DFT results in the main body of the manuscript employ implicit solvation, and gas phase results have been moved to the supporting information.

I also didn't quite understand why the x-ray experiments were deemed to be "very instructive." At one level, and in my opinion on reading the new theory section, the conformation thrown up by the x-ray experiments could be seen to be confusing or artifactual. The authors mentioned that the observed conformation is unexpected and concluded that it is due to "molecular ordering." It was not immediately obvious to me what "molecular ordering" meant in this context, since molecular ordering occurs in both the solid state and in solution; I assume that they are referring to crystal packing? The authors could rewrite this making more explicit reference to crystal packing or solid state as opposed to molecular ordering and also highlighting that the crystal structure conformation is likely to be a solid state artifact once seen in the context of NMR experiments and DFT calculations, which is what I concluded. Furthermore, now considering that this conformation may be due to crystal packing and considering that they have access to the unit cell do they understand and can they explain why? This would also help convince the reader.

In the manuscript, we changed "molecular ordering" for "crystal packing".

DFT calculations help us to understand why our fluorinated sugar is in the GG conformation in the solid state. Using DFT, we determined that:

- 1) the GG conformer is the least favorable staggered conformers;
- 2) the dipole moment of the GG conformer is the highest of the three staggered conformers (this high dipole moment might explain favorable molecular crystal packing of polarized pyran rings);
- 3) the GG conformer benefits the most when analysed in small clusters of repeat units (see supporting information);
- 4) NBO analysis ruled out the possibility of hydrogen bonding involving fluorine atoms.

We hope the reader will follow this line of thought to better understand why the GG conformer is found in the solid state. We thank the Reviewer for their concern and suggestions as it has led to a clearer understanding of the results.

The authors hope that this revised manuscript fulfills all the requested criteria for *Nature Communications*.

Reviewer #4 (Remarks to the Author):

My main concern was around solvent effects. This has been addressed and indeed does appear to have an effect, which has been reported.

The manuscript is now acceptable for publication.